# Effects of Acute Hyperthermia on the Thermotolerance of Cow and Sheep Skin-Derived Fibroblasts

**DOI:** 10.3390/ani10040545

**Published:** 2020-03-25

**Authors:** Islam M. Saadeldin, Ayman Abdel-Aziz Swelum, Adel M. Zakri, Hammed A. Tukur, Abdullah N. Alowaimer

**Affiliations:** 1Department of Animal Production, College of Food and Agricultural Sciences, King Saud University, 11451 Riyadh, Saudi Arabia; aswelum@ksu.edu.sa (A.A.-A.S.); tukurhammeda@gmail.com (H.A.T.); aowaimer@ksu.edu.sa (A.N.A.); 2Department of Physiology, Faculty of Veterinary Medicine, Zagazig University, 44519 Zagazig, Egypt; 3Department of Theriogenology, Faculty of Veterinary Medicine, Zagazig University, 44519 Zagazig, Egypt; 4Plant Protection Department, College of Food and Agricultural Sciences, King Saud University, 11451 Riyadh, Saudi Arabia; azakri@ksu.edu.sa

**Keywords:** fibroblasts, hyperthermia, thermotolerance, heat shock proteins, cows, sheep

## Abstract

**Simple Summary:**

We compared the thermotolerance of cow and sheep fibroblasts after exposure to acute hyperthermia (45 °C for 4 h). The primary culture, first passage, and cryopreserved cow fibroblasts resisted acute hyperthermia in terms of cell viability, proliferation, and migration to close cell scratch, in addition to increased expression of heat shock protein (HSP70 and HSP90) mRNA transcripts.

**Abstract:**

This study was conducted to compare the effects of acute hyperthermia (45 °C for 4 h) on the viability, proliferation, and migratory activity through wound-healing assays of cow and sheep fibroblasts. The study examined the effects on primary cultures and first passage skin-derived fibroblasts. Relative quantification of HSP70, HSP90, P53, BAX, BCL2, and BECN1 was investigated after normalization to housekeeping genes GAPDH and beta-actin. The results revealed that cultured cow primary fibroblasts exhibited increased viability and reinitiated cell migration to close the cell monolayer scratch earlier than sheep cells. Similar patterns were observed in the first passage fibroblasts, with severe effects on sheep cells. Both cow and sheep cells exhibited decreased cell viability and failed to regain migratory activity after re-exposure of recovered heat-shocked cells. Effects of hyperthermia on sheep cells were potentiated by cell cryopreservation. The qPCR results showed that cow cells significantly increased HSP70 and HSP90 expression, which decreased the elevation of P53, and ameliorated the effects of the increased BAX/BCL2 ratio. The results provide a paradigm to compare thermotolerance among different animal species and revealed that trypsin could be an additional stress, which potentiates the effects of heat shock in in vitro experiments.

## 1. Introduction

Heat stress (HS) has major effects on milk production and fertility in cows and sheep [1,2,3,4,5,6]. Genetic variations in susceptibility to hyperthermia have been reported among different species [7]. Acclimation or adaptation to increased temperatures basically depends on physical mechanisms, such as heat dissipation through conduction, convection, radiation, and evaporation by means of sweating or panting [8,9,10]. At the cellular level, acclimation can occur as part of the cellular heat tolerance mechanisms. In cattle and sheep cells, thermotolerance is higher in breeds adapted to warm climates than in animals with temperate origins [11,12,13,14]. This resistance could be mediated by heat shock protein (HSP)-related mechanisms [15]. HSPs are a family of proteins with diverse molecular weights (100, 90, 70, and 60 kDa) and act to prevent thermal damage to proteins [16,17]. This acute response plays a pivotal role in the cellular thermal acclimation mechanism to high temperatures and could be used as a marker for the adaptation capabilities of animals to thermal stress [8,16,18,19].

Among HSPs, HSP70, and HSP90 are the two major types that have the potential to bind to unfolded proteins, helping them to fold and synthesize properly [20,21]. Some types of these proteins could be expressed substantially, whereas most of them are expressed under stress conditions [22]. Several environmental or physiologic stresses could lead to the production and activation of HSPs, such as inflammation, hypoxia, chemotherapy, and infections, as well as thermal injury [3,16,17,23,24]. In living systems, HSPs have essential activities, including polypeptide folding, protein transportation, and multiprotein complex formation [25]. Moreover, they can prevent apoptosis, clear aggregated proteins, and ameliorate the cytotoxic impact of toxic proteins.

Thermotolerance is a phenomenon in which cells become resistant to elevated temperatures. Thermotolerance can develop rapidly after the first heat treatment or during thermal treatment at ~43.0 °C. Studies have shown that thermotolerance also develops in tumors and normal tissues [26,27] and is well correlated with the enhanced synthesis of heat shock proteins [27,28,29,30]. The kinetics of thermotolerance can be affected by various factors [26,31]. For instance, thermotolerance was found physiologically in certain species as a form of estivation [32,33]. It could vary among certain cells in the same species [17,34]. Cells show variability in thermotolerance, apoptosis, and necrosis depending on the cell culture method; the three-dimensional (3D) culture system shows increased survival plausibly by activating the protective processes related to enhanced HSP70 expression as compared with the two-dimensional (2D) culture system at 45 °C for 120 min [35].

Recently, we reported the thermotolerance of camel somatic cells to acute and chronic thermal stress, as evidenced by increased HSP expression, antioxidative response, and DNA repair enzymes [16]. Additionally, our previous results showed that camel cells tolerated acute hyperthermia and showed no alteration in the apoptosis-related mRNA transcripts P53, BAX, and BCL2); a proteomic investigation revealed the response of the cells with increased protein levels associated with autophagy to adapt to such lethal conditions [16].

Because of the essential role of HSPs in thermotolerance, we and other researchers proposed the phenomenon of “anastasis” to illustrate the survival response of thermotolerant cells [36,37,38,39,40]. Anastasis is a process of plasticity, resilience, or cellular resurrection from the brink of cell death [39] and could explain the difference in the thermotolerance of cells [41].

Additionally, there has been no reports regarding the differences between sheep and cow cell thermotolerance in vitro, which could reflect the acclimation, adaptation, or cellular response of the whole body to HS. Therefore, this study aimed to compare the effects of acute HS on cow and sheep skin-derived fibroblasts to provide a paradigm to elucidate the adaptability of large and small ruminant species to in vitro hyperthermia.

## 2. Materials and Methods

### 2.1. Chemicals

Chemicals were purchased from Sigma-Aldrich Corp. (St. Louis, MO, USA) unless otherwise stated.

### 2.2. Skin Tissue Sampling and Culture

Skin tissue samples were obtained from a local abattoir after five healthy cows (Holstein breed) and sheep (Naimi breed) were slaughtered. The whole ears of cows and sheep were transferred to the laboratory within 3 h, cleaned with povidone iodide and 70% ethanol, and shaved to remove the hair. Skin samples from the ear cutaneous pouch were chopped into small 0.5 cm^2^ pieces and soaked in sterile Dulbecco’s phosphate buffered saline (DPBS) with a high concentration (10×) of antibiotic/antimycotic (penicillin (500 IU/mL0, streptomycin (500 µg/mL), and amphotericin (2.5 µg/mL)).

Skin-derived fibroblasts were isolated and cultured according to our previous protocols [42,43]. In brief, skin tissue pieces were transferred to 60 mm tissue culture plates using a sterile scalpel. The culture medium comprised DMEM (cat no. D6171, Sigma) containing glucose (4.5 g/L), HEPES (25 mM), L-glutamine (0.584 g/L), 10% fetal bovine serum (FBS; Gibco, Waltham, MA, USA), and a 10× penicillin, streptomycin, and amphotericin solution and incubated at 37 °C in an atmosphere of 5% CO_2_. The cultures were monitored microscopically to observe explant attachment and outgrowth of primary fibroblasts on day 10 of the culture. To maintain the primary cell culture, we removed the remaining parts of the skin explants and excluded the heterogeneous or mixed cell types according to our previous report [24].

### 2.3. Cryopreservation and Thawing of Fibroblasts

Cow and sheep primary cells were trypsinized with 0.25% trypsin-EDTA for 2 min at 37 °C and cells were centrifuged at 200× *g* for 2 min. The pellets were resuspended in freezing medium (50% DMEM, 10% DMSO, and 40% FBS (v/v)). Cell suspensions were distributed into cryogenic vials (at approximately 1 × 10^6^ cells/vial), gradually frozen at −80 °C, and transferred to a liquid nitrogen tank for long-term storage, until they were used for further experiments. Immediately before experiments, vials were quickly thawed at 37 °C, promptly mixed with the culture medium, and cultured in 60 mm culture dishes.

### 2.4. Measuring Cellular Attachment and Cell Viability

Images of cells (at least 10 fields) were captured pre- and post-exposure and analyzed using ImageJ 1.50i software (NIH, Bethesda, USA), after exposure was ended and cells were washed with PBS. To calculate the proportion of attached and non-attached cells, the percentage of attached cells in post-exposure to the initial cells, pre-exposure, was estimated. For the viability examination, cells were first trypsinized and the cell suspension was mixed at a 1:1 ratio with 0.4% trypan blue solution and incubated for 1 min at room temperature. Then, the cell suspension was loaded into a hemocytometer chamber and cells were counted to assess their viability using a traditional cell-counting method [17].

### 2.5. Assessment of Cell Migration through Wound Healing Assay

A wound healing assay was conducted according to our previous protocol [17,24]. Briefly, confluent cell monolayers were scraped into a straight line with a 200 μL yellow pipette tip to create a wound. Cell debris and non-adherent cells were removed by washing with 2 mL of DMEM. Serial digital images of the wound were captured and the wound width was measured using ImageJ 1.50i software (NIH, Bethesda, USA) at different time points as referred to by the automated scale bar of the inverted microscope (Leica DMI4000 B; Leica Microsystems GMS GmbH, Wetzlar, Germany) for pixel analysis. The culture medium comprised DMEM, 10% FBS, and a 1× penicillin/streptomycin/amphotericin solution. After creating the wound, cells were exposed to specific heat exposure as described below.

### 2.6. Experimental Design

Cell cultures and heat treatments were performed in 6-well culture plates after cell plating, attachment, and attaining 70% to 80% confluency, and one well of each dish was used to evaluate the temporal cellular viability. Each experiment was repeated five times.

#### 2.6.1. Effect of Acute Heat Shock on Primary Fibroblast Viability and Migratory Activity

Cow and sheep primary cultured cells were exposed to acute heat shock (45 °C for 4 h), then the medium was changed with a fresh culture medium and cultured at 38 °C for recovery from heat shock. The cell architecture was microscopically checked, and the cell viability and wound healing were compared against the control fibroblasts cultured at 38 °C for 4 h.

#### 2.6.2. Effects of Acute Heat Shock on the Recovered Primary Fibroblast Viability and Migratory Activity

The primary fibroblasts that recovered and reached 70% to 80% confluency were exposed to acute heat shock (45 °C for 4 h), and then the medium was changed and cultured at 38 °C for recovery. Cells were microscopically checked for architecture and growth. Cell viability and wound healing were recorded and compared to the control fibroblasts cultured at 38 °C for 4 h.

#### 2.6.3. Impacts of Acute Heat Shock on the First Passage Fibroblast Viability and Migratory Activity

Trypsinized and subcultured fibroblast cells were exposed to acute heat shock at 45 °C for 4 h.

#### 2.6.4. Impact of Acute Heat Shock on the Cryopreserved Fibroblasts Viability and Migratory Activity

Cryopreserved fibroblasts were resuscitated and exposed to acute heat shock at 45 °C for 4 h.

In Appendix A, we provide an illustration of the experimental design.

### 2.7. Relative Quantitative Polymerase Chain Reaction (qPCR)

Total RNA extraction was isolated through a commercial kit (SV Total RNA Isolation System; Promega, Madison, WI, USA) following the manufacturer’s instructions. Cultured cells (three replicates) were lysed with lysis buffer that was provided with the kit. The RNA quantity and quality were evaluated using a NanoDrop 2000 spectrophotometer (Thermo Fisher). Reverse transcription (RT) was performed according to our previous report [16], with 120 cycles of 16 °C for 2 min, 37 °C for 1 min, and 50 °C for 1 s, followed by a final inactivation at 85 °C for 5 min. RT reactions comprised 50 ng of total RNA, and 5 µM of random hexamers in a 20 µL total reaction volume using an Amfivert cDNA Synthesis Master Mix (GenDEPOT Inc., TX, USA). Relative quantitative real-time PCR was performed using an automated thermal cycler (ViiA 7; Applied Biosystems). Reactions comprised 100 ng of cDNA, 1 µM forward and reverse primers, and 1× SYBR Green Premix (Applied Biosystems). Housekeeping genes *GAPDH* and *ACTB* were used for normalization, and the fold changes of the target transcripts (HSP70, HSP90, P53, BAX, BCL2, and BECN1) were calculated using the 2^−ΔΔCt^ method [24]. In all assays, reactions without cDNA and without RT resulted in no amplification. Thermal cycling conditions were 95 °C for 5 min, followed by 40 cycles of 95 °C for 10 s, 60 °C for 30 s, and 72 °C for 30 s. Information on primers is listed in Appendix A.

### 2.8. Statistical Analysis

Three to five replicates were reported for each examined parameter. Cell viability data were presented as means ± SEM, and values were analyzed using an ANOVA followed by Tukey’s post-hoc test. Data on wound width, cellular attachment, and qPCR were presented as means ± SEM, and values were analyzed using Student’s *t*-test. Differences were considered significant at *p* < 0.05. Pearson’s linear correlation coefficients were calculated to determine the correlation (*R*) between the means of different mRNA transcript expressions in the primary culture and first passage fibroblasts individually for cow and sheep cells, where *R* > ±0.7 were considered strong positive/negative linear relationships; *R* > ±0.5, moderate positive/negative linear relationships; and *R* < ±0.5, weak positive/negative linear relationships [42].

## 3. Results

### 3.1. Effects of Acute Heat Shock on the Viability of Primary Fibroblasts in Cows and Sheep

The primary culture of fibroblasts exhibited different behavior for recovery when exposed to acute heat shock. Cow cells regained proliferation and reached confluence after 96 h of recovery (Figure 1); however, sheep cells reached confluence after 192 h of recovery at 38 °C (Figure 1 show the cell proliferation until 96 h, although we did not show the entire results to be consistent in the species comparison). Interestingly, cells of the first passage in cows showed thermotolerance and regained the proliferation and confluency after 240 h of recovery, whereas sheep cells showed signs of degeneration and cells were easily removed during washing and changing of the culture medium (Appendix A). In contrast, primary cultured cells that recovered from the initial exposure showed cell degeneration and did not reach confluency, even after 240 h of recovery (Figure 1).

### 3.2. Effects of Acute Heat Shock on the Migratory Activity of Primary Fibroblasts in Cows and Sheep

Figure 2 and Figure 3 represent the comparison between the tolerance of primary cultured fibroblasts in cows and sheep when exposed to acute heat shock (45 °C for 4 h). Cow fibroblasts showed rapid wound closure (cell scratch assay) as compared with the sheep fibroblasts, 96 h vs. 192 h, respectively. The high magnification of the cells showed that sheep cells lost the attachment to the culture dish at a relatively higher proportion than that of the cow cells (Appendix A).

### 3.3. Effects of Acute Heat Shock on the Migratory Activity of the Recovered Primary Fibroblasts in Cows and Sheep

Figure 4 and Figure 5 represent the comparison between the tolerance of the primary cultured fibroblasts in cows and sheep that were recovered from previous exposure and re-exposed to acute heat shock (45 °C for 4 h). Both cells showed degeneration and failed to close the wounds even after 240 h of recovery. However, high magnification of the cells showed that sheep cells lost their attachment to the culture dish at a relatively higher proportion as compared with that of the cow cells (Appendix A), but with the progression of time, the cell viability decreased and the cells were easily detached during routine washing and medium changes.

### 3.4. Effects of Acute Heat Shock on the Migratory Activity of the First Passage Cow and Sheep Fibroblasts

Figure 6 and Figure 7 represent the comparison between the tolerance of the first passage cow and sheep fibroblasts exposed to acute heat shock (45 °C for 4 h). Cow fibroblasts showed wound closure after 240 h from recovery at 38 °C, whereas sheep fibroblasts failed to attain wound healing at this time and showed degenerative changes. High magnification of the cells showed that sheep cells lost their attachment to the culture dish at a relatively higher proportion than did cow cells (Appendix A).

### 3.5. Effects of Acute Heat Shock on the Viability of Cryopreserved Fibroblasts in Cows and Sheep

Cryopreserved cow cells showed better thermotolerance than did sheep cells exposed to acute heat shock (45 °C for 2 h). Viable cells dropped to 1.7 ± 0.5 vs. 7.1 ± 0.7 × 10^4^/mL as compared with the control cells at 10.2 ± 0.5 vs. 10.6 ± 0.6 × 10^4^/mL in sheep and cows, respectively (*p* < 0.05) (Figure 8). Additionally, cow cells showed an increased proportion of attachment as compared with sheep cells (53.4 ± 3.6% vs. 16.6 ± 4.3%, respectively, *p* < 0.05) (Figure 8).

### 3.6. Effects of Acute Heat Shock on the Expression of mRNA Transcripts in Primary Culture and First Passage Cow and Sheep Fibroblasts

In both cow and sheep primary cultured fibroblasts (Figure 9A,B), HSP70 and HSP90 showed increased expression in heat-shocked cells as compared with the control cells. However, BECN1 showed significant decreases in both cell types. Paradoxically, BAX and BAX/BCL2 showed significant increases in cow primary cultured fibroblasts as compared with that in sheep cells. BCL2 exhibited no change in sheep cells, whereas cow cells showed a decreased level. P53 showed a significant increase in sheep cells as compared with that in cow and control cells. In the first passage (Figure 9C,D), cow cells still showed the increased expression of both HSP70 and HSP90, whereas sheep cells showed the increased expression of only HSP90. P53 exhibited more than a 16-fold increase in sheep cells as compared with the lack of change in cow cells. BAX and BAX/BCL2 showed a significant increase in both cell types, whereas BECN1 exhibited a significant decrease in cow cells. BCL2 exhibited significant decreases in both cells. In cow cells (Appendix A), HSP90 expression had a strong positive correlation with HSP70, BAX, and BAX/BCL2 expression, whereas it showed a strong negative correlation with P53, BCL2, and BECN1. Similarly, HSP70 expression had a strong negative correlation with BCL2 and BECN1 and a moderate positive correlation with BAX/BCL2. Conversely, P53 had a strong positive correlation with BCL2, a strong negative correlation with BAX and BAX/BCL2, and a moderate positive correlation with BECN1. In contrast, BAX showed a strong negative correlation with BCL2 and a strong negative correlation with BECN1. In sheep cells, HSP90 had a strong positive correlation with P53, a moderate positive correlation with BAX, and a moderate negative correlation with BCL2. However, HSP70 had a moderate negative correlation with BAX and a strong negative correlation with BECN1. In addition, P53 exhibited a strong positive correlation with BAX and BECN1 and a strong negative correlation with BCL2 (Appendix A). 

## 4. Discussion

The current results represent a continuation of our experiments that were performed to compare the species differences in in vitro thermotolerance to elucidate species adaptability and acclimation to extreme heat exposure [16,17,24,34]. Recently, we showed that camel cells better tolerated chronic heat shock (45 °C for 20 h) than porcine cells [16], and the results of this study revealed that cow cells were more tolerant to acute heat shock than sheep cells on the primary and first passage level, as well as for cryopreserved fibroblasts. Cell viability, proliferation, attachment to culture dishes, and ability to close the wound were all significantly decreased in sheep cells as compared with in cow cells.

To investigate these mechanisms, we measured the fold changes of some mRNA transcripts associated with heat shock response (HSP70, HSP90), apoptosis (P53, BAX, and BCL2), and autophagy (BECN1).

It is difficult to compare the species differences in terms of gene expression; however, in the current approach, we measured the relative fold changes in the expression of the same mRNA transcripts within the same species and compared its expression pattern with those of the other species, in accordance with a recent report [43].

Although the BAX/BCL2 ratio was elevated in cow fibroblasts, the P53 transcript expression was significantly elevated in both the primary culture and first passage fibroblasts of sheep cells as compared with that in cow cells. Correlation analysis showed that P53 exhibited a strong positive correlation with apoptosis (BAX) and autophagy (BECN1)-related mRNAs. However, there was a strong positive correlation of P53 with HSP90 in sheep as compared with a strong negative correlation in cows. This result indicated that cow cells responded with HSP90 and HSP70 in a greater proportion than did sheep cells.

In bovine cells, it has been reported that hyperthermia caused somatic cell apoptosis because of an imbalance in the BAX/BCL2 pathway in favor of the pro-apoptotic protein BAX [44,45]. Moreover, stress signals can trigger the increase in P53 expression for cell cycle arrest, DNA repair, and apoptosis [46]. Conversely, the co-increase in the expression of HSP90 and HSP70 in the first passage cow fibroblasts could be the reason behind the relative thermotolerance of cow cells as compared with that of sheep fibroblasts in the first passage (Figure 9C and Appendix A). HSP70 inhibits apoptosis by blocking the recruitment of procaspase-9 to the cytochrome c apoptosome complex through conformational changes that render procaspase-9 binding less effective [47]. Therefore, P53 can be protected from thermal unfolding by HSPs, and this protection can require reciprocal interactions with different HSP domains [48,49]. It is apparent from the current study that the elevated P53 expression could reduce cell viability and thermotolernace in sheep cells and this could be a result of the relatively high expression of both HSP70 and HSP90 in cow cells as compared with in sheep cells. Additionally, P53 showed a moderate and strong positive correlation with the autophagy gene (BECN1) in cow and sheep cells, respectively, which indicated that the activation of autophagy is a part of the negative feedback and protective function of P53 [50].

Surprisingly, cow and sheep cells that were recovered from the heat shock and re-exposed to similar conditions lost their coping capabilities and resistance to heat shock, which indicated that acclimation or adaptation is not likely to be achieved in vitro under our experimental conditions.

Species and breed differences in thermotolerance have been reported in both sheep and cows. The cell viability in the Pelibuey breed was higher than that in the Suffolk breed after hyperthermia, an effect that could be attributed to an HSP-70-related mechanism [15]. In contrast, thermotolerance and HSP90 expression were elevated in breeds adapted to warm climates as compared with breeds from colder origins, such as Sahiwal (*Bos indicus*) and Frieswal (*Bos indicus* × *Bos taurus*) [13], or between Zebu and crossbreed cattle [14], respectively.

On the basis of the current results, we confirm our previous findings that trypsin could be an additional stress to the cells in vitro [24]. Furthermore, exposure of cells to trypsin potentiated the effects of heat on cellular damage [51,52]. Furthermore, trypsin has been found to alter cell membrane surface proteins that are responsible for the extracellular matrix and metabolic pathways [53,54]. Trypsin is the most common enzyme used for cell passaging and routine cell culture protocols; hence, it is important to find an alternative for cell dissociation in experiments studying cellular responses to heat shock. Moreover, cryopreservation showed the reduced thermotolerance of sheep cells, which could be an additional stress to the cells. Cryopreservation has been proven to be a successful approach for the long-term storage of biological material; however, there is growing evidence of its potentially damaging effects at the molecular level of cells, such as metabolic perturbations, impaired mitochondrial function, and gene expression [55,56,57]. The effects of cryopreservation could also be potentiated by initial cell dispersion with trypsin, which causes additional stress to the cells.

## 5. Conclusions

This study investigated the thermotolerance of cow and sheep fibroblasts to acute hyperthermia (45 °C for 4 h). Cow cells showed greater resistance than sheep cells in terms of cell viability, proliferation, and migratory activities, as well as the maintenance of HSP90 and HSP70 expression to overcome the damaging effects of hyperthermia. However, further studies are necessary to analyze the evolutionary and molecular mechanisms controlling different gene and protein expressions associated with thermotolerance in these types of cells.

## Figures and Tables

**Figure 1 animals-10-00545-f001:**
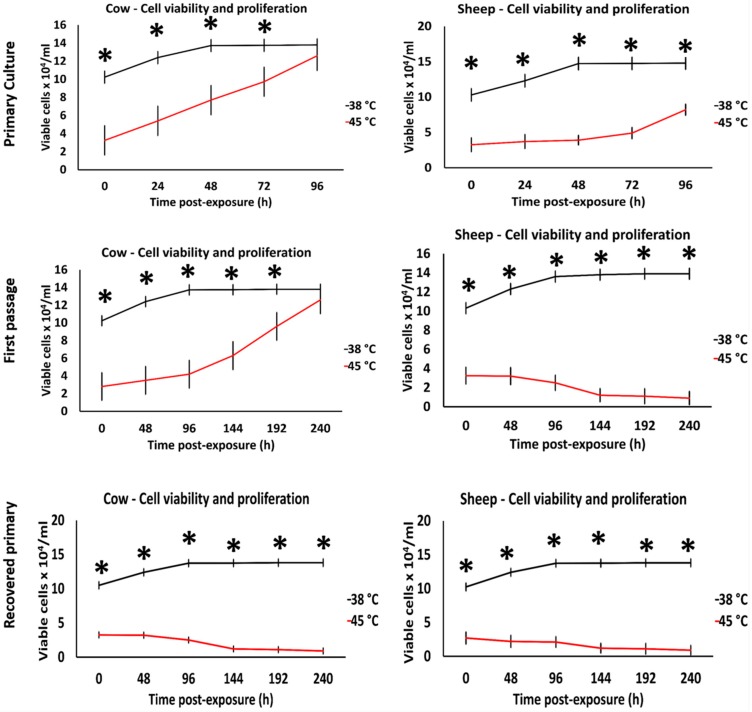
Effects of acute heat stress (45 °C for 4 h) on cow and sheep cell viability. After the end of exposure (0 h), cells were recovered at 38 °C in a humid atmosphere of 5% CO_2_ and examined daily for cell viability until they reached 100% confluency (~13 × 10^4^ cells/mL). Data are presented as means ± SEM, values of the same timings were analyzed using Student’s *t*-test, and asterisks (*) indicate significant differences at *p* < 0.05.

**Figure 2 animals-10-00545-f002:**
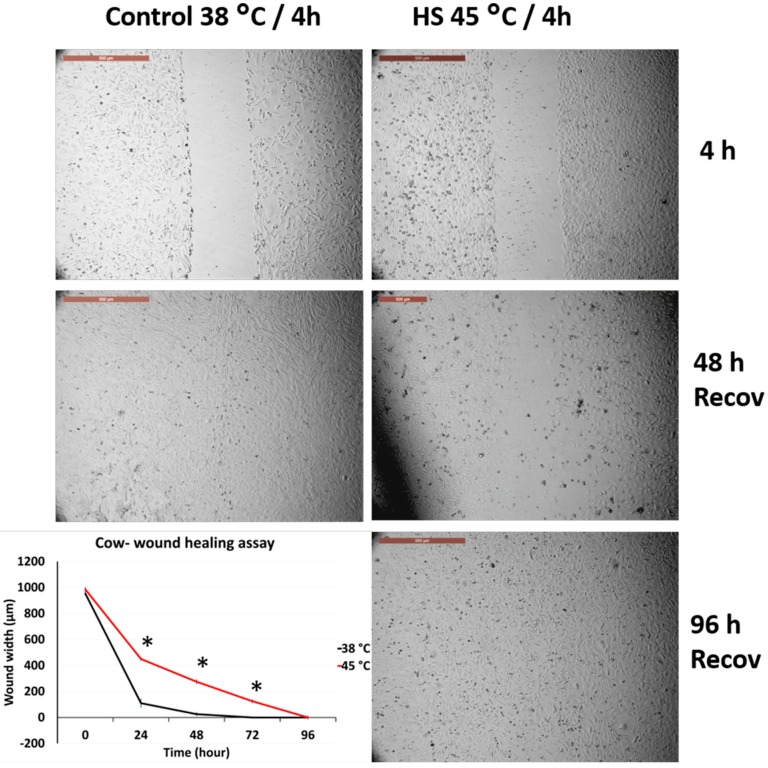
Wound healing (cell scratch) assay for cow primary cultured fibroblasts exposed to 45 °C for 4 h and control cells (38 °C for 4 h). Cells were incubated under the required conditions, recovered at 38 °C, and monitored daily for wound closure. Scale bar = 500 µm. Data are presented as means ± SEM, values of the same timings were analyzed using Student’s *t*-test, and asterisks (*) indicate significant differences at *p* < 0.05.

**Figure 3 animals-10-00545-f003:**
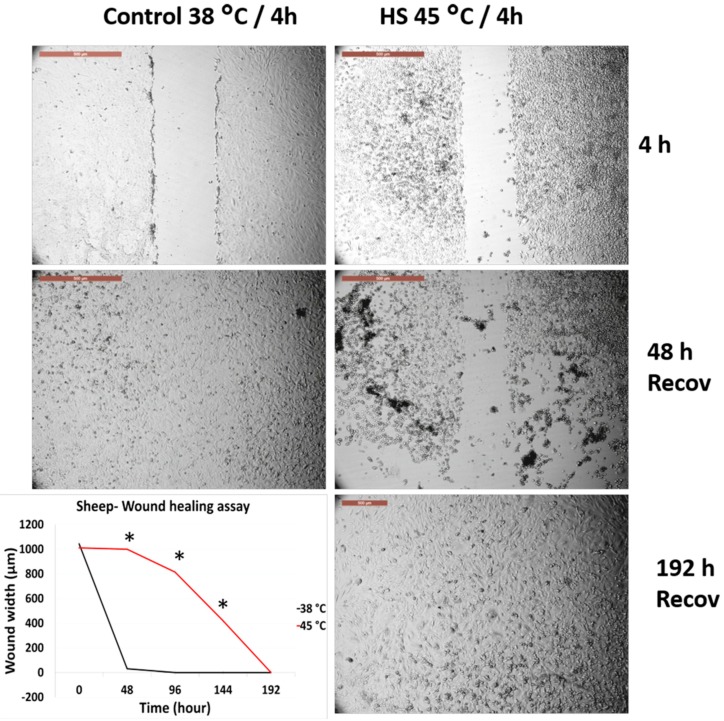
Wound healing (cell scratch) assay for sheep primary cultured fibroblasts exposed to 45 °C for 4 h and control cells (38 °C for 4 h). Cells were incubated under the required conditions, recovered at 38 °C, and monitored daily for wound closure. Scale bar = 500 µm. Data are presented as means ± SEM, values of the same timings were analyzed using Student’s *t*-test, and asterisks (*) indicate significant differences at *p* < 0.05.

**Figure 4 animals-10-00545-f004:**
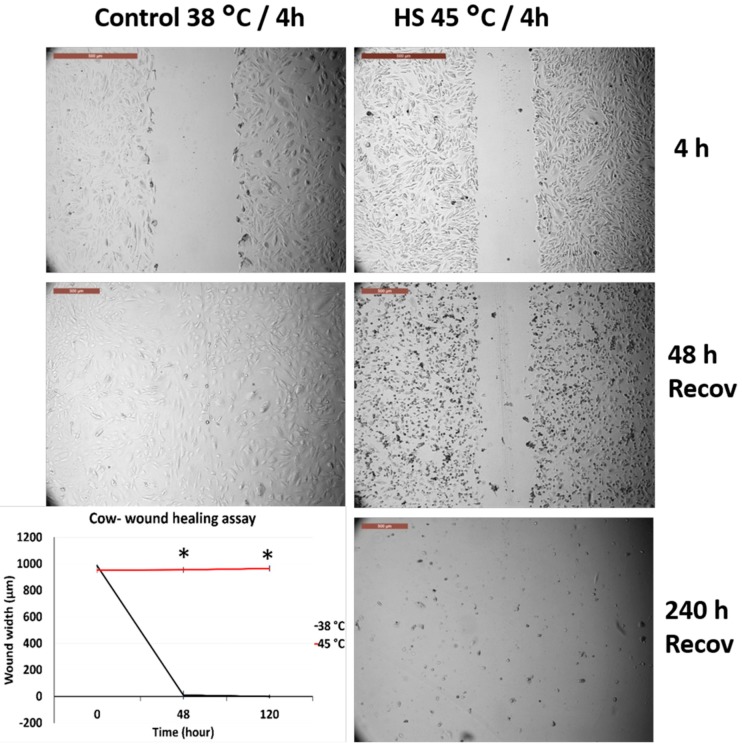
Wound healing (cell scratch) assay for cow primary cultured fibroblasts that recovered after exposure to 45 °C for 4 h and reached confluency as compared with the control cells (38 °C for 4 h). The wounds were created after cell recovery at confluency. Cells were incubated under the required conditions, recovered at 38 °C, and monitored daily for wound closure. Scale bar = 500 µm. Data are presented as means ± SEM, values of the same timings were analyzed using Student’s *t*-test, and asterisks (*) indicate significant differences at *p* < 0.05.

**Figure 5 animals-10-00545-f005:**
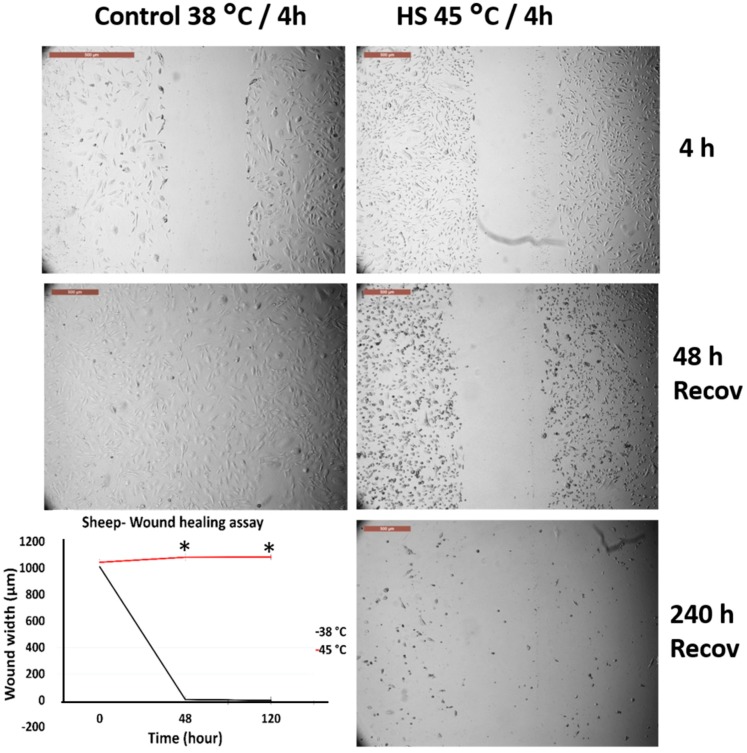
Wound healing (cell scratch) assay for sheep primary cultured fibroblasts that recovered after exposure to 45 °C for 4 h and reached confluency as compared with the control cells (38 °C for 4 h). The wounds were created after cell recovery at confluency. Cells were incubated under the required conditions, recovered at 38 °C, and monitored daily for wound closure. Scale bar = 500 µm. Data are presented as means ± SEM, values of the same timings were analyzed using Student’s *t*-test, and asterisks (*) indicate significant differences at *p* < 0.05.

**Figure 6 animals-10-00545-f006:**
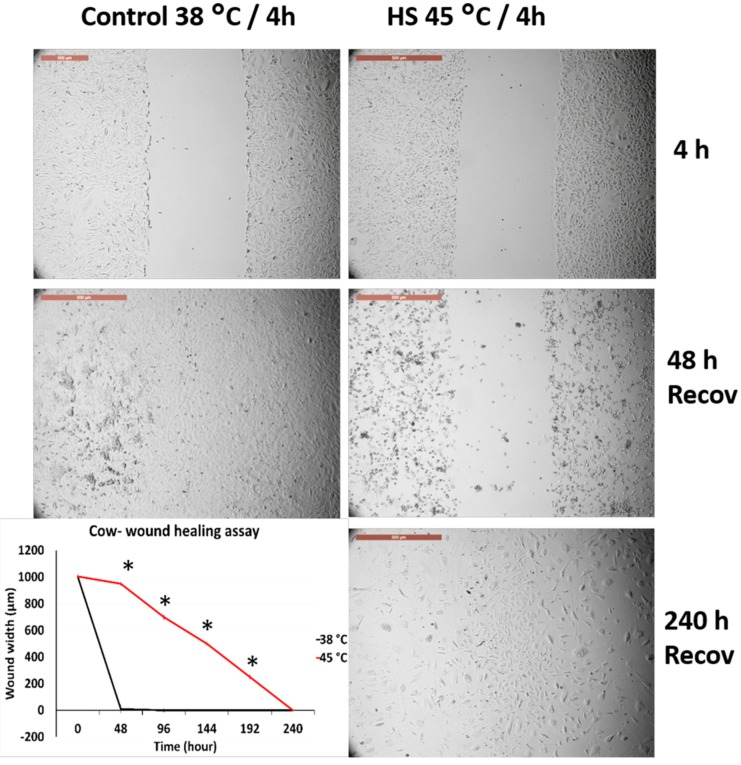
Wound healing (cell scratch) assay for cow first passage fibroblasts exposed to 45 °C for 4 h and control cells (38 °C for 4 h). Cells were incubated under the required conditions, recovered at 38 °C, and monitored daily for wound closure. Scale bar = 500 µm. Data are presented as means ± SEM, values of the same timings were analyzed using Student’s *t*-test, and asterisks (*) indicate significant differences at *p* < 0.05.

**Figure 7 animals-10-00545-f007:**
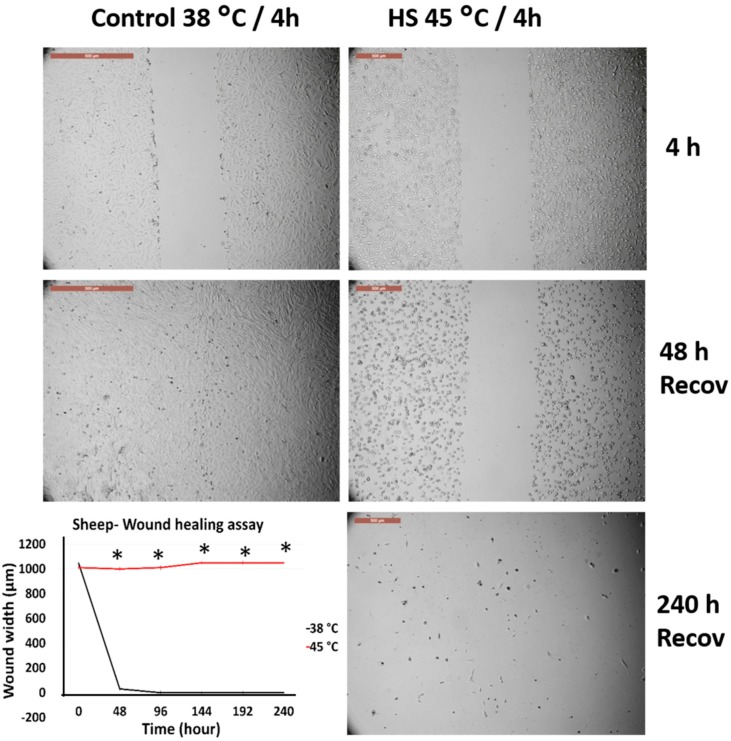
Wound healing (cell scratch) assay for sheep first passage fibroblasts exposed to 45 °C for 4 h and control cells (38 °C for 4 h). Cells were incubated under the required conditions, recovered at 38 °C, and monitored daily for wound closure. Scale bar = 500 µm. Data are presented as means ± SEM, values of the same timings were analyzed using Student’s *t*-test, and asterisks (*) indicate significant differences at *p* < 0.05.

**Figure 8 animals-10-00545-f008:**
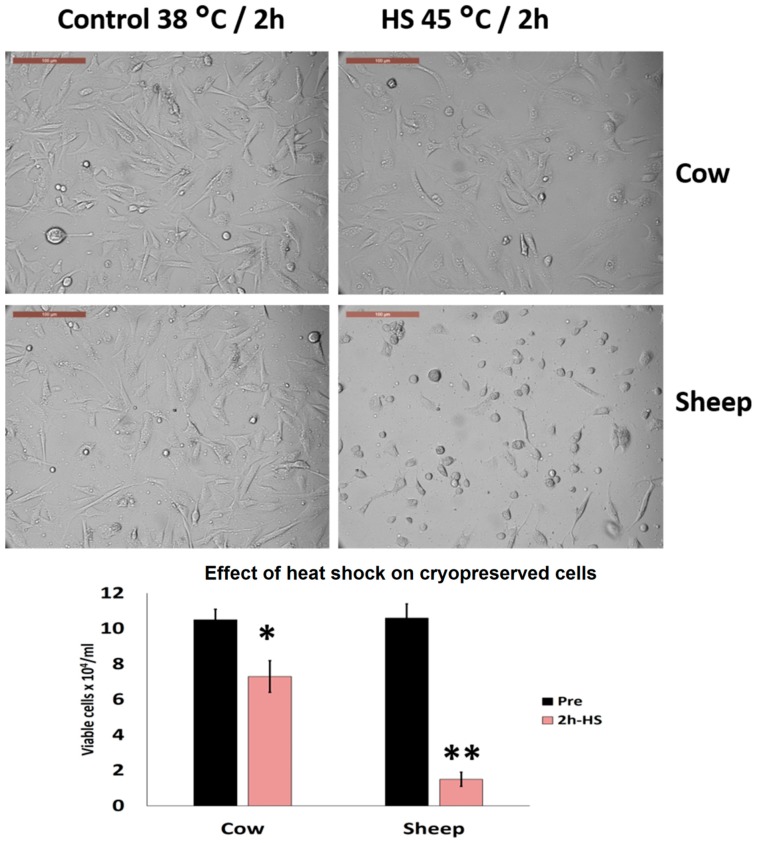
High magnification of cow and sheep cryopreserved fibroblasts after exposure to acute heat shock (4 °C for 2 h). In cow cells, several cells remained attached to the cell culture dish and retained a spindle morphology (arrows), whereas, in sheep, the majority of the cells lost their architecture (arrows) and attachment to the culture dish. The numbers of attached and non-attached cells were counted and analyzed with ImageJ software and the data were analyzed by Student’s *t*-test. Scale bar = 100 µm. The graph describes the effects on cell viability, data are presented as means ± SEM, values of the same timings were analyzed using a Student’s *t*-test, and asterisks (*) and (**) indicate significant changes at *p* < 0.05 and *p* < 0.01, respectively.

**Figure 9 animals-10-00545-f009:**
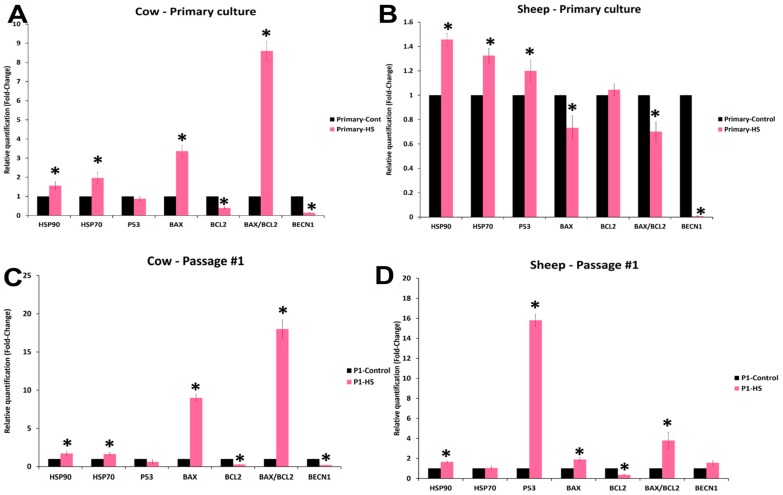
Relative quantitative PCR and fold change of selected mRNA transcript expressions in cow and sheep fibroblasts of different passages exposed to acute heat shock (45 °C for 4 h). (**A**,**B**) show the primary cultured fibroblasts in cows and sheep, respectively; (**C**,**D**)show the first passage (P1) fibroblasts in cows and sheep, respectively. Data are presented as means ± SEM, values of the same mRNA transcript in control and heat-shocked cells were analyzed using Student’s *t*-test, and asterisks (*) indicate significant differences at *p* < 0.05.

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
