# Peer review of "Effects of Acute Hyperthermia on the Thermotolerance of Cow and Sheep Skin-Derived Fibroblasts"

_animals, 2020, doi:10.3390/ani10040545_

Round 1
Reviewer 1 Report
The opening statement of the Abstract claims: “This study was conducted to compare effects of acute lethal hyperthermia (45°C for 4 hours)” What is the basis that exposure to 45°C for 4 hours is lethal? No evidence is provided to support this unjustified claim.
The opening paragraphs of the Introduction are well written and provide a sound basis to justify the study. From the style of the Introduction I detect the following sections were not written by the same author. The statement (ln 70) “…cells grown in 3D compared to 2D cultures exhibit a reduced incidence of apoptosis and necrosis…” is an over-generalization and demonstrably false. Evidence from studies using other cell types do not support this claim (see e.g. Physiology (Bethesda). 2017; 32: 266–277; Oncotarget. 2017; 8:107423-107440; Sci Rep 8: 3627 (2018)) In order to permit others to reproduce this study it would be helpful to know from which anatomical region donor skin was harvested. Skin from the flanks is unlike skin from the limbs, is unlike skin from the head, is unlike skin from the ears. Precisely, what is (ln 93) “…a high concentration (10X) of antibiotic/antimycotic…”? (ln 97) Which formulation of DMEM was used (i.e. 4500 mg/L glucose, or 1000 mg/L glucose)? Glucose has a significant impact on mammalian cell physiology and functional activity. How can cell attachment / non-attachment – a functional parameter – be determined from (2 dimensional) images? The use of 0.4% Trypan Blue to measure cell viability is widely reported to yield a high incidence of false positives. Exclusion of Trypan Blue is not a robust measure for cell viability. (See Biol Proced Online. 2017; 19: 8) The relative age (i.e. duration in culture) of mammalian cells grown in monolayers has a significant influence on their detachment, where by enzymatic or mechanical means. That is, monolayer cultivars that have secreted substantial ECM (i.e. aged) are considerably more difficult to ‘scrape-wound’ than are monolayer cultivars with little secreted ECM. This is especially a challenge when working with epithelial cells, rather than mesenchymal cells as described in this manuscript, but remains a critical (unreported) parameter. While creating scratch wounds using pipette tips is economical and convenient, are the authors aware that pipette tips also cause substantial damage to culture vessel surfaces? Not only are cells removed, so too are attachment factors and ECM glycoproteins that facilitate cell anchorage. Pipette tips also impose mechanical damage to vessel surfaces, creating 3 dimensional physical features that distort cellular responses. Primary (i.e. freshly isolated) cells cultivated in vitro do not constitute a ‘cell line’, as claimed in line 130. It is not at all clear precisely when (ln 130) “primary cells were exposed to acute heat shock”. Were cells heat shocked before plating? after plating? or after attachment? What protocol do I use to reproduce this experiment? A critical aspect of maintaining mammalian cells in vitro (and ex vivo, as described in this report) is pH. I am sure that the authors are well aware that pH is highly temperature sensitive. What measures did the authors take to ensure cell culture media pH did not fluctuate significantly? They should also be aware regulated cell death pathways other than apoptosis (e.g. alkaliptosis, acidosis, oxeiptosis) are temperature-sensitive. Osmolality is also affected by temperature. How were (ln 135) “…fibroblasts that recovered…” identified? (ln 146) What is “lysis buffer”? (ln 158) What is “template-negative samples”? I interpret that this is laboratory jargon; it is not a term with which this reader is familiar. (ln 179) It is claimed that “cells were easily removed during washing and changing of culture media” is illustrated in Fig 1. Figure 1 does not illustrate this. Figure 1 illustrates “Effects of acute heat stress”. All figure legends are inadequate. No description of the basic method is provided. No indication of what asterisks mean, the whiskers represent, or the applied statistical test is reported. In Figure 1, each ordinate is labelled as “Viable cells x 104/mL”. Given that the reported cells grow as adherent monolayers, how is it possible to report viability as a concentration – per mL? Figure 1. Why are cell cultures exposed to treatment at 45°C have a population at time = 0, <20% cell populations exposed to 38°C? For accurate comparison, shouldn’t each population start with the same density?What statistical test was applied to determine statistical significance? (ln 189) No consideration is given to measures of basal cell motility (i.e. random motility). Surely this also offers a measure of cell responses to thermal stress? (ln 191) Supplemental Figure 1 does not report the loss of cell attachment!
How were attached and non-attached cells determined? Into which category are the mitotic indices evident in the images, assigned? (ln 194) Figure 2 illustrates images captured from a scratch wound assay. It is evident that after 4 hours the wound includes many cells that were displaced during scratching and since have reattached to the wounded area. These cells should not be evident; they corrupt the data and illustrate poor technique.
Also illustrated in Figure 2 is evidence for edge effects (right middle image annotated 48 h Recov (sic)).
Why does the ordinate for wound width include a measure increment of “–200 µm”?
What statistical test was applied to determine statistical significance? (ln 199) Figure 3 is annotated as “Wound healing assay for cow primary… fibroblasts” It reports, in fact, data for sheep fibroblasts.
What statistical test was applied to determine statistical significance? (ln 212) Figure 4 reports “Wound healing assay for cow primary… fibroblasts… after 45°C…” The reduction in wound width (1 mm) for control samples requires 48 hours. This is unusually slow and not what this reviewer would anticipate observing. Something is not right here.
What statistical test was applied to determine statistical significance? (ln 218) Why is the top left image reported at a different magnification to all other images in this figure?
What statistical test was applied to determine statistical significance? (ln 251) What do the arrows in the lower right-hand image represent?
How was cell viability determined? In my opinion the data reported in Figure 9 “Relative quantitative PCR…” are overinterpretted. Changes in mRNA quantitation in the order of 20% - 40% are statically and functionally meaningless
What statistical test was applied to determine statistical significance?
(ln293) Where is the data reporting “reporting migratory activities” described in the text. (ln 304) Where is the “Correlation analysis” described in the text. (ln 525) Supplemental Figure 3. How were attached and non-attached cells determined? How were attached and non-attached cells counted?Author Response
General response: We appreciate the time, efforts, and overall constructive suggestions and comments raised by the reviewer that greatly contributed to the improvement of manuscript quality. All the suggestions have been addressed, editing and corrections made accordingly. To give a brief description for the work, in general, we used the same conditions to culture cow and sheep fibroblasts (same timing, same handling, same incubator, same media, even the same person who handled both cells). In all experiments, as we mentioned in the protocols, we repeated three to five times to be statistically valid.
- The opening statement of the Abstract claims: “This study was conducted to compare effects of acute lethal hyperthermia (45°C for 4 hours)” What is the basis that exposure to 45°C for 4 hours is lethal? No evidence is provided to support this unjustified claim.
R1- We edited the term into “acute hyperthermia”. We depended on our previous results in camel (Saadeldin et al. 2019 Environ Sci Pollut Res 26, 29490–29496, Doi: 10.1007/s11356-019-06208-5), however in agreement with the reviewer point of view, we modified it in the title and throughout the manuscript.
- The opening paragraphs of the Introduction are well written and provide a sound basis to justify the study.
R2- Thanks for this comment.
- From the style of the Introduction I detect the following sections were not written by the same author. The statement (ln 70) “…cells grown in 3D compared to 2D cultures exhibit a reduced incidence of apoptosis and necrosis…” is an over-generalization and demonstrably false. Evidence from studies using other cell types do not support this claim (see e.g. Physiology (Bethesda). 2017; 32: 266–277; Oncotarget. 2017; 8:107423-107440; Sci Rep 8: 3627 (2018))
R3- We disagree with the reviewer, we quoted these findings from the cited reference and here is the statement as written in the abstract of Ref [37] Song et al. 2014 “Cells grown in 3D compared with 2D culture showed reduced incidence of apoptosis and necrosis and a higher level of HSP70 expression in response to heat shock at the temperatures tested. Cells responded differently to hyperthermia when grown in 2D and 3D cultures. Three-dimensional culture appears to enhance survival plausibly by activating protective processes related to enhanced-HSP70 expression. These differences highlight the importance of selecting physiologically relevant 3D models in assessing cellular responses to hyperthermia in experimental settings’. In fact this finding supports our claim in terms of heat shock. Therefore, we modified the text to be more specific.
- In order to permit others to reproduce this study it would be helpful to know from which anatomical region donor skin was harvested. Skin from the flanks is unlike skin from the limbs, is unlike skin from the head, is unlike skin from the ears.
R4- Clarified and we provided an additional illustration for more clarity.
- Precisely, what is (ln 93) “…a high concentration (10X) of antibiotic/antimycotic…”? (ln 97) Which formulation of DMEM was used (i.e. 4500 mg/L glucose, or 1000 mg/L glucose)? Glucose has a significant impact on mammalian cell physiology and functional activity.
R5- Clarified.
- How can cell attachment / non-attachment – a functional parameter – be determined from (2 dimensional) images?
R6- This parameter is known for all persons handling cell culture and it well known in the literature that viable cells are able to attach to attach to the culture dish (polystyrene materials) and this was discussed early in 1965 by Priest et al. (Cell attachment assay as a measure of thawed cell viability. Cryobiology Vol 1, Issue 5, May–June 1965, Pages 345-347). Moreover, we showed the results of attachment in the Supplemental Figures (S2, S3, and S4).
For more confirmation, the parameter of attachment or confluency can be calculated through automated cell analyzer as a proportion of confluency (We used this method before in South Korea at our recent report; see Saadeldin et al. 2020 PMID). However, through using our current capability, we performed it manually through the same concept using ImageJ program’s cell counter property.
- The use of 0.4% Trypan Blue to measure cell viability is widely reported to yield a high incidence of false positives. Exclusion of Trypan Blue is not a robust measure for cell viability. (See Biol Proced Online. 2017; 19: 8).
R7- Although this was the available test to examine cell viability and we have an additional confirmatory test which is the migration or scratch healing assay that yielded a matching result with Trypan Blue, so we disagree with the reviewer because the provided Ref. of Piccinini et al. (Biol Proced Online. 2017; 19: 8) supports the use of Trypan Blue for assessing cell viability but not for cell density because there is expected variability between the practitioners “They stated that: The results obtained prove that (a) there is no significant difference between 2D and 3D cell cultures as far as TB reliability is concerned; (b) the TB method is precise when used for viability assessments of a cell culture; (c) the method is fairly inaccurate at estimating cell population density, despite it is routinely used for this purpose in numerous laboratories”. We hope our response clear.
- The relative age (i.e. duration in culture) of mammalian cells grown in monolayers has a significant influence on their detachment, where by enzymatic or mechanical means. That is, monolayer cultivars that have secreted substantial ECM (i.e. aged) are considerably more difficult to ‘scrape-wound’ than are monolayer cultivars with little secreted ECM. This is especially a challenge when working with epithelial cells, rather than mesenchymal cells as described in this manuscript, but remains a critical (unreported) parameter. While creating scratch wounds using pipette tips is economical and convenient, are the authors aware that pipette tips also cause substantial damage to culture vessel surfaces? Not only are cells removed, so too are attachment factors and ECM glycoproteins that facilitate cell anchorage. Pipette tips also impose mechanical damage to vessel surfaces, creating 3 dimensional physical features that distort cellular responses.
R8- We thank the reviewer for this notion. We agree with the reviewer for this point however, we previously handled and reported the scratch techniques, the cells are doing well particularly the control cells, and there was no observed effects on their reaching to confluency.
- Primary (i.e. freshly isolated) cells cultivated in vitrodo not constitute a ‘cell line’, as claimed in line 130. It is not at all clear precisely when (ln 130) “primary cells were exposed to acute heat shock”. Were cells heat shocked before plating? after plating? or after attachment? What protocol do I use to reproduce this experiment?
R9- We clarified the points.
- A critical aspect of maintaining mammalian cells in vitro (and ex vivo, as described in this report) is pH. I am sure that the authors are well aware that pH is highly temperature sensitive. What measures did the authors take to ensure cell culture media pH did not fluctuate significantly?
R10- You are right for this point, and according to our experiments, we used HEPES-supplemented DMEM to minimize the effects of pH. However, this point is now under experimental investigation by our group, we sent the cells and the culture media to another collaborator laboratory to analyze the metabolomics and to explore the possible altered metabolic pathways for both species.
It is known that HEPES is widely used in cell culture, largely because it is better at maintaining physiological pH despite changes in carbon dioxide concentration (produced by aerobic respiration) when compared to bicarbonate buffers, which are also commonly used in cell culture.
- They should also be aware regulated cell death pathways other than apoptosis (e.g. alkaliptosis, acidosis, oxeiptosis) are temperature-sensitive. Osmolality is also affected by temperature.
R11- We agree with the reviewer but we selected these pathways in the current study and considered other pathways for our further investigations of these interesting pathways.
- How were (ln 135) “…fibroblasts that recovered…” identified?
R12- They reached the required confluency after recovery from heat shock.
- (ln 146) What is “lysis buffer”?
R13- The lysis buffer that was prvided with kit (SV Total RNA Isolation System, Promega) and supplemented with β-mercaptoethanol as recommended by the manufacturer.
- (ln 158) What is “template-negative samples”? I interpret that this is laboratory jargon; it is not a term with which this reader is familiar.
R14- We modified the text for easy understanding.
- (ln 179) It is claimed that “cells were easily removed during washing and changing of culture media” is illustrated in Fig 1. Figure 1 does not illustrate this. Figure 1 illustrates “Effects of acute heat stress”.
R15- We are sorry for this mistyping, we corrected it into Supplemental Fig. S2.
- All figure legends are inadequate. No description of the basic method is provided. No indication of what asterisks mean, the whiskers represent, or the applied statistical test is reported.
R16- We thank the reviewer for this note, we corrected the legends accordingly.
- In Figure 1, each ordinate is labelled as “Viable cells x 104/mL”. Given that the reported cells grow as adherent monolayers, how is it possible to report viability as a concentration – per mL? Figure 1. Why are cell cultures exposed to treatment at 45°C have a population at time = 0, <20% cell populations exposed to 38°C? For accurate comparison, shouldn’t each population start with the same density? What statistical test was applied to determine statistical significance? (ln 189) No consideration is given to measures of basal cell motility (i.e. random motility). Surely this also offers a measure of cell responses to thermal stress? (ln 191)
R17- We responded and clarified the viability in our previous answer to Q7. For the cells at 0h, this count is expected because the 0h counts after ending the exposure period and we showed that the cell viability reduced because of heat shock. In addition, we edited the figure legend to show the statistical method. For motility, we also showed our response before. We also would like to draw the attention of the reviewer that these parameters were reported by our group since we started the series of work on heat shock on camel cells (please see Saadeldin et al. 2020 PMID: 31969994; Saadeldin et al. 2019 PMID: 31435907 PMID: 31226810; and Saadeldin et al. 2018 PMID: 29801649. In addition, this experiment is a continuation of our work but on cow and sheep cells. We depend on cell morphology, wound healing assay, cell viability, and time-lapse videos, as well as the mRNA transcript expression and proteomics to understand the full story.
- Supplemental Figure 1 does not report the loss of cell attachment!
How were attached and non-attached cells determined? Into which category are the mitotic indices evident in the images, assigned?
R18- The proportion of cell attachments was already clarified in methodology at L115.
- (ln 194) Figure 2 illustrates images captured from a scratch wound assay. It is evident that after 4 hours the wound includes many cells that were displaced during scratching and since have reattached to the wounded area. These cells should not be evident; they corrupt the data and illustrate poor technique.
R19- Actually, we were careful to place the dishes in recovery after washing two times with PBS and culture medium and the appeared cells in the site of the wound as dark died cells or cellular fragments which were not attached to the culture dish even after recovery and these cells were removed by the routine medium change every 48 h.
Also illustrated in Figure 2 is evidence for edge effects (right middle image annotated 48 h Recov (sic)).
R- This is an artifact because of the marker used to label the culture dishes.
- Why does the ordinate for wound width include a measure increment of “–200 µm”?
What statistical test was applied to determine statistical significance? (ln 199)
R- We clarified it in the legend.
- Figure 3 is annotated as “Wound healing assay for cow primary… fibroblasts” It reports, in fact, data for sheep fibroblasts.
What statistical test was applied to determine statistical significance? (ln 212)
R20- We apologize for this mistake. We corrected it. Moreover, we clarified figure legend.
- Figure 4 reports “Wound healing assay for cow primary… fibroblasts… after 45°C…” The reduction in wound width (1 mm) for control samples requires 48 hours. This is unusually slow and not what this reviewer would anticipate observing. Something is not right here.
R- No, that is we have found for all control cells in Figures 2-8. The observation of wound healing at 24 h was not completely healed however after 48 h was completely healed. It might be attained healing before <48h or >24h, but to be consistent with the timing, we fixed the observations at daily basis.
- What statistical test was applied to determine statistical significance? (ln 218)
R- We clarified it in the legend.
- Why is the top left image reported at a different magnification to all other images in this figure?
R- This was the shape of the cells after recovery. Please note that the cells were kept for around 15 days in the culture dish and this might change the cell size. You will find that Fig. 4 and 5 with a similar cell morphology in sheep and cows.
- What statistical test was applied to determine statistical significance?
R- We showed the method of cell attachment judgment in our previous response.
- (ln 251) What do the arrows in the lower right-hand image represent?
R- We are sorry, we clarified it Figure 8 legend.
How was cell viability determined?
R- We described it before.
- In my opinion the data reported in Figure 9 “Relative quantitative PCR…” are overinterpretted. Changes in mRNA quantitation in the order of 20% - 40% are statically and functionally meaningless. What statistical test was applied to determine statistical significance?
R22- We clarified it in the legend. These results are from 3 biological and 3 technical replicates.
- (ln293) Where is the data reporting “reporting migratory activities” described in the text.
R23- The term was deleted.
- (ln 304) Where is the “Correlation analysis” described in the text.
R24- This was described in the methods and the data are in the Supplementary file.
- (ln 525) Supplemental Figure 3. How were attached and non-attached cells determined? How were attached and non-attached cells counted?
R25- We answered it be previously.

Reviewer 2 Report
The paper presents a comprehensive study that provide grounds to compare heat stress differential effects on cow and sheep fibroblasts.
Some minor suggestions are:
More clarity in the description of successive experiments would be appreciated in order to know the differences between “Primary Culture”, “First passage” and “Recovered primary”. A graph showing the transition between stages might help.
Please, add some justification in the introduction for the” use of expression of P53, BAX, BCL2, and BECN1 genes to monitor cell status along the experiment.
Also, why were fibroblasts were chosen to do the experiment instead of other may be more interesting cells from a productive perspective such as germ cells?
Figure1. Please use same scales (range of values in y axis) and size in all graphs to improve visual comparison between cow and sheep results. Also, in the title, “daily” should be every two days? Results are shown for 48h apart intervals. Moreover, the meaning of the asterisks is not provided in the figure caption.
Figure 3. In the figure’s caption “cow” should be “sheep”.
Figure 9. As for Figure 1, I suggest to use same scales (range of values in y axis) and size in all graphs to improve visual comparison between cow and sheep results.
Conclusion. The first sentence is not a real conclusion derived from the results of the experiments
In the supplementary tables of fold change correlations, how did you decide what is moderate and what a high correlations? I guess that this was done arbitrarily. If so, I’d rather let the reader take their own conclusion about magnitude of correlations. Also, I do not think that the clarification about positive or negative correlation is needed. Readers of a scientific paper should know what a correlation is and how to interpret it.
Author Response
The paper presents a comprehensive study that provide grounds to compare heat stress differential effects on cow and sheep fibroblasts.
Response: We acknowledge the time, efforts, and comments by the reviewer. We substantially revised the manuscript according your suggestion.
Some minor suggestions are:
- More clarity in the description of successive experiments would be appreciated in order to know the differences between “Primary Culture”, “First passage” and “Recovered primary”. A graph showing the transition between stages might help.
R1- Thank you for the suggestion, we provided an illustration for the experimental design to be more clear (Supplemental Fig. S1).
- Please, add some justification in the introduction for the” use of expression of P53, BAX, BCL2, and BECN1 genes to monitor cell status along the experiment.
R3- We thank the reviewer for the useful suggestion, we added a paragraph describing them.
- Also, why were fibroblasts were chosen to do the experiment instead of other may be more interesting cells from a productive perspective such as germ cells?
R3- We thank the reviewer for this comment. We choose fibroblast as it is easily isolated from skin culture, however, we tried the effects on oocytes (i.e. as a model for germ cells) in camels (see our report Saadeldin et al. 2018 J Thermal Biol, PMID: 29801649, DOI:
10.1016/j.jtherbio.2018.03.014). We are planning to study the effects on cows, sheep, and goat as models of tropical animals.
- Please use same scales (range of values in y axis) and size in all graphs to improve visual comparison between cow and sheep results. Also, in the title, “daily” should be every two days? Results are shown for 48h apart intervals. Moreover, the meaning of the asterisks is not provided in the figure caption.
R4- We thank you for this notion, we corrected the figures’ captions. Regarding the X-axis range, we were adhered to the confluency of the cells, so some cells reached at 96 h and hence the range was stopped at 96h and showed 24h apart. On the other hand, some cells reached at 192h or 240 h, so we lengthened the x-axis and jumped the range to 48h or more. However, we monitored and measured the wound diameter daily in our experiment but we expressed the range either in 24h, 48h, or even more according to the target, which is the confluency as mentioned above. We hope that you kindly consider our point.
- Figure 3. In the figure’s caption “cow” should be “sheep”.
R5- We apologize for this mistake. We corrected it.
- Figure 9. As for Figure 1, I suggest to use same scales (range of values in y axis) and size in all graphs to improve visual comparison between cow and sheep results.
R6- We tried to do this, but because there is some mRNAs that are around 20-fold differences, so arbitrary group (1-fold) will be very small to appear. So, we kept the figure unmodified. We added the explanation for the asterisk and adjusted the size as possible.
- The first sentence is not a real conclusion derived from the results of the experiments
R7- We edited the text.
- In the supplementary tables of fold change correlations, how did you decide what is moderate and what a high correlations? I guess that this was done arbitrarily. If so, I’d rather let the reader take their own conclusion about magnitude of correlations. Also, I do not think that the clarification about positive or negative correlation is needed. Readers of a scientific paper should know what a correlation is and how to interpret it.
R8- we thank the reviewer for this comment. We clarified the method of correlation in the statistical analysis (section 2.9.) and gave the appropriate reference for that. In fact, it is not a fold-change; it is a matter of correlation coefficient and based on the P value of the statistical test as stated in the cited reference.

Reviewer 3 Report
This study compared thermotolerance of cow and sheep fibroblasts to acute
lethal temperature (45 °C for 4 h). Though this study provides some evidence that cow cells showed greater resistance than sheep cells in terms of cell viability and higher expression of HSP90 and HSP70 to overcome the damaging effects of hyperthermia, there is a limited practical significance of the results of this study which should ideally look at the systemic responses (thermoregulatory mechanisms) of the animals not just cells.
Please check the simple summary again for some spelling mistakes in Line 21.
Author Response
Response: We appreciate the time, efforts, and suggestions by the reviewer. All the suggestions have been addressed, and corrections made accordingly.
- This study compared thermotolerance of cow and sheep fibroblasts to acute lethal temperature (45 °C for 4 h). Though this study provides some evidence that cow cells showed greater resistance than sheep cells in terms of cell viability and higher expression of HSP90 and HSP70 to overcome the damaging effects of hyperthermia, there is a limited practical significance of the results of this study which should ideally look at the systemic responses (thermoregulatory mechanisms) of the animals not just cells.
R1- We thank the reviewer for the excellent reviewing and for the useful suggestion. We are looking to apply a controlled environment for the live animal study in collaboration with the partner laboratory in the near future.
- Please check the simple summary again for some spelling mistakes in Line 21.
R2- We thank the reviewer for the comment. We corrected the mistyping in the simple summary.

Round 2
Reviewer 1 Report
The manuscript, “Effects of acute hyperthermia on the thermotolerance of cow and sheep skin-derived fibroblasts” describes in vitro experiments designed to test the hypothesis that dermal fibroblasts derived from bovine and ovine donors are sensitive to heat stress, that the responses of donor-derived dermal fibroblasts is genetically pre-determined and likely to indicate the capacity of each species to tolerate (and adapt to) environmental heat stress. The unspecified implication is how in vitro data might predict tolerance for climate change.
My main criteria for accepting/rejecting draft manuscripts is reproducibility: can another who is practiced in the art, reproduce the experiments and obtain equivalent results using only the provided description. For this manuscript, the answer is unlikely.
The incidence of self-citations (8/60) is higher than I usually observe.
The authors have responded in detail to my first review.
However, I find that some of their responses demonstrate they have not understand the scientific basis for several of my comments.
- (R3) My original comment: “…cells grown in 3D compared to 2D cultures exhibit a reduced incidence of apoptosis and necrosis…” is an over-generalization and demonstrably false is roundly disputed by the author. Fair enough; however, surely our divergence of opinion represented an opportunity for debate and discussion. I note no evidence in their discussion of the cause(s) of cell death in the revised manuscript. Thus, my comment and supporting evidence is a lost opportunity for debate!
- (R6) In my expert opinion the authors fail to comprehend my observation that 2-dimensional images are not a reliable measure of cell attachment, or of non-attachment. Cell attachment is a physical parameter; it must be determined using a physical measure! 2-dimensional images do not provide such a physical measure.
- (R7) I accept the authors argument regarding their application of Trypan Blue; however, their rationale is spurious. It is not good science. I suggest that they familiarise themselves with the wider literature and cease to rely on Trypan Blue as a measure of cell viability. I remain sceptical.
- (R8) While accepting my comments, the authors compound poor reporting, claiming “we previously handled and reported the scratch techniques, the cells are doing well (sic) particularly the control cells, and there was no observed effects on their reaching to confluency” (sic). While I am a practicing laboratory scientist, I am not able to replicate “doing well”, nor can I deduce what measurable parameter “observed effects” refers to?
My question remains unaddressed.
- (R12) “reaching the required confluency after recovery from heat shock” is not an accurate method to identify any specific cell population. What is the evidence that recovered cells were indeed ‘fibroblasts’, and not a contaminating sub-population of other origin?
- (R17) I remain to be convinced that comparing data acquired from different cell population densities after exposure to heat-shock is a valid experimental strategy.
The fact that previous reviewers have overlooked this weak scientific methodology is not reason to accept it now! I maintain that this experimental protocol is not a valid approach to evaluate responses of different cell populations exposed to traumatic stress. It is recognised that cell-cell contact and communication is a critical parameter to maintain the integrity and responses of multicellular and heterogeneous tissues. Thus, sub-populations of cells challenged at different densities will respond uniquely based on their density, rather than their inherent resistance, or sensitivity, to the challenge (in this example heat stress). - (R18) My question remains unaddressed.
Author Response
The manuscript, “Effects of acute hyperthermia on the thermotolerance of cow and sheep skin-derived fibroblasts” describes in vitro experiments designed to test the hypothesis that dermal fibroblasts derived from bovine and ovine donors are sensitive to heat stress, that the responses of donor-derived dermal fibroblasts is genetically pre-determined and likely to indicate the capacity of each species to tolerate (and adapt to) environmental heat stress. The unspecified implication is how in vitro data might predict tolerance for climate change.
Response: Thank you for your excellent review that greatly contributed to improving the quality of the manuscript. We have clarified this in the first paragraph of the introduction.
My main criteria for accepting/rejecting draft manuscripts is reproducibility: can another who is practiced in the art, reproduce the experiments and obtain equivalent results using only the provided description. For this manuscript, the answer is unlikely.
The incidence of self-citations (8/60) is higher than I usually observe.
We are sorry for the mistake of Endnote software, three references were repeated unintentionally (Ref 34 is originally the 17th), (Ref 36 and Ref 44 are originally for the 24th). Therefore, they have been removed from the References’ list, and hence the list has been updated.
The authors have responded in detail to my first review.
Thank you for your excellent review that greatly contributed to improving the quality of the manuscript.
However, I find that some of their responses demonstrate they have not understand the scientific basis for several of my comments.
- (R3) My original comment: “…cells grown in 3D compared to 2D cultures exhibit a reduced incidence of apoptosis and necrosis…” is an over-generalization and demonstrably false is roundly disputed by the author. Fair enough; however, surely our divergence of opinion represented an opportunity for debate and discussion. I note no evidence in their discussion of the cause(s) of cell death in the revised manuscript. Thus, my comment and supporting evidence is a lost opportunity for debate!
R3- We have modified the text to show the reason for cell death, which were apoptosis and necrosis caused by the imbalance between caspase and HSP70 response when cells were exposed to 45 C for 2 h.
- (R6) In my expert opinion the authors fail to comprehend my observation that 2-dimensional images are not a reliable measure of cell attachment, or of non-attachment. Cell attachment is a physical parameter; it must be determined using a physical measure! 2-dimensional images do not provide such a physical measure.
R6- We previously showed this our previous response and here we confirm that the parameter of attachment or confluency can be calculated through automated cell analyzer as a proportion of confluency (We used this method before in South Korea at our recent report; [Time-lapse imaging was performed using JuLITM Live Cell Movie analyzer (NanoEnTek Inc., Seoul, Korea], see Saadeldin et al. 2020 PMID 31969994: https://www.ncbi.nlm.nih.gov/pubmed/31969994). However, through using our current capability in our lab, we performed it manually through the same concept using ImageJ program’s cell counter property. The concept of analyzing 2D culture uses image analysis as the following:
Haenel F, Garbow N. Cell Counting and Confluency Analysis as Quality Controls in Cell-Based Assays. Multimode Detection. 2014:1-5.
Busschots et al. Non-invasive and non-destructive measurements of confluence in cultured adherent cell lines. MethodsX 2015, 2: 8-13.
Schor SL, Court J. Different mechanisms in the attachment of cells to native and denatured collagen. J Cell Sci. 1979, 38:267-81.
And this typically relates to our manual method for cell counting and judgment of cell attachment in the 2D culture system.
- (R7) I accept the authors argument regarding their application of Trypan Blue; however, their rationale is spurious. It is not good science. I suggest that they familiarise themselves with the wider literature and cease to rely on Trypan Blue as a measure of cell viability. I remain sceptical.
R7: We agree with the reviewer and we will consider the advanced tools such as Annexin v/PI staining and flow cytometry for our future experiments.
- (R8) While accepting my comments, the authors compound poor reporting, claiming “we previously handled and reported the scratch techniques, the cells are doing well (sic) particularly the control cells, and there was no observed effects on their reaching to confluency” (sic). While I am a practicing laboratory scientist, I am not able to replicate “doing well”, nor can I deduce what measurable parameter “observed effects” refers to?
My question remains unaddressed.
R8- We are sorry for this misunderstanding. We meant that all cells were cultured at the same time with the control cells and our analysis was based on the cell behavior and comparison against the control as an arbitrary group.
- (R12) “reaching the required confluency after recovery from heat shock” is not an accurate method to identify any specific cell population. What is the evidence that recovered cells were indeed ‘fibroblasts’, and not a contaminating sub-population of other origin?
R12- We previously showed that the isolated cells were fibroblasts and provided a supplemental figure to show this origin, that based on previous findings too.
- (R17) I remain to be convinced that comparing data acquired from different cell population densities after exposure to heat-shock is a valid experimental strategy.
The fact that previous reviewers have overlooked this weak scientific methodology is not a reason to accept it now! I maintain that this experimental protocol is not a valid approach to evaluate responses of different cell populations exposed to traumatic stress. It is recognised that cell-cell contact and communication is a critical parameter to maintain the integrity and responses of multicellular and heterogeneous tissues. Thus, sub-populations of cells challenged at different densities will respond uniquely based on their density, rather than their inherent resistance, or sensitivity, to the challenge (in this example heat stress).
R17- We agree with the reviewer for this point, but as we mentioned previously, we run the experiments against the control cells that were in the same passage and the same initial cell number or cell density.
- (R18) My question remains unaddressed.
R18- the original question was “Supplemental Figure 1 does not report the loss of cell attachment!
How were attached and non-attached cells determined? Into which category are the mitotic indices evident in the images, assigned? And our answer was “The proportion of cell attachments was already clarified in methodology at L115” which is clear that we have shown the method for cell attachment calculation. For the mitotic index, we have not measured the mitotic index but we are planning to do so in our future experiments.